# The Monkey Head Mushroom and Memory Enhancement in Alzheimer’s Disease

**DOI:** 10.3390/cells11152284

**Published:** 2022-07-24

**Authors:** Wing Shan Yu, Man Lung Fung, Chi Wai Lee, Lee Wei Lim, Kah Hui Wong

**Affiliations:** 1School of Biomedical Sciences, Li Ka Shing Faculty of Medicine, The University of Hong Kong, Hong Kong, China; yanshree@connect.hku.hk (Y.); yuwsw@connect.hku.hk (W.S.Y.); fungml@hku.hk (M.L.F.); chiwai.lee@hku.hk (C.W.L.); 2Department of Anatomy, Faculty of Medicine, Universiti Malaya, Kuala Lumpur 50603, Malaysia

**Keywords:** *Hericium erinaceus*, Alzheimer’s disease, aging, memory, preclinical, clinical

## Abstract

Alzheimer’s disease (AD) is a neurodegenerative disorder, and no effective treatments are available to treat this disorder. Therefore, researchers have been investigating *Hericium erinaceus*, or the monkey head mushroom, an edible medicinal mushroom, as a possible treatment for AD. In this narrative review, we evaluated six preclinical and three clinical studies of the therapeutic effects of *Hericium erinaceus* on AD. Preclinical trials have successfully demonstrated that extracts and bioactive compounds of *Hericium erinaceus* have potential beneficial effects in ameliorating cognitive functioning and behavioral deficits in animal models of AD. A limited number of clinical studies have been conducted and several clinical trials are ongoing, which have thus far shown analogous outcomes to the preclinical studies. Nonetheless, future research on *Hericium erinaceus* needs to focus on elucidating the specific neuroprotective mechanisms and the target sites in AD. Additionally, standardized treatment parameters and universal regulatory systems need to be established to further ensure treatment safety and efficacy. In conclusion, *Hericium erinaceus* has therapeutic potential and may facilitate memory enhancement in patients with AD.

## 1. Introduction

Neurodegenerative disorders are an umbrella term for a wide range of conditions. Alzheimer’s disease (AD) constitutes nearly 50–70% of all neurodegenerative disorder-related illnesses for dementia [1]. With the increasing older population worldwide, it is predicted that over 130 million individuals over the age of 65 will develop AD by 2050 [2]. Alzheimer’s disease causes considerable social and economic burdens, especially for patients, their families, and caregivers. Accumulation of amyloid-beta (Aβ) plaques and hyperphosphorylation of tau leading to neurofibrillary tangles (NFTs) are characteristic hallmarks of AD, resulting in key symptoms such as memory loss, difficulty in recalling recent incidents, and overall behavior changes [3,4,5]. In AD, the symptoms can range from mild to severe. In mild condition, AD patients have a greater tendency to misplace items and show poor judgment. As AD progresses, individuals can have difficulty identifying their family members, cannot learn new information, and lose the ability to categorize thoughts. Individuals with severe AD can lose the ability to communicate and are prone to infections and chronic inflammation, eventually leading to death [1]. Early diagnosis and treatment are important, as progression of the disease leads to irreversible cognitive decline.

Although there are some treatments currently available for AD, the multifactorial and complex nature of this disease have led to questions about their effectiveness [6]. Current AD therapeutics primarily focus on alleviating the symptoms and slowing down the disease progression [7,8]. Acetylcholinesterase inhibitors (AChEl) including donepezil and galantamine and memantine (N-methyl-D-aspartic acid (NMDA) receptor antagonists) can potentially treat the cognitive decline in AD [9]. It has been shown that AChEls can increase transmission of acetylcholine in the brain including the cerebral cortex via preventing acetylcholine breakdown by cholinesterase in synapses [10], whereas NMDA receptor antagonists can protect neuronal cells against excitotoxicity due to excessive activation of NMDA receptors [11]. Monoclonal antibodies such as Solanezumab and Aducanumab target aggregated Aβ peptides to remove excess Aβ plaques [10,12]. Several potential approaches for blocking tau hyperphosphorylation have also been developed [13]. Despite these current treatments, new and safer drugs are needed, as patients on AD medications often experience adverse side effects [14,15]. In addition, these medications only inhibit the symptoms of AD for a limited period and have low efficacy in decelerating the progressive deterioration. Further research should focus on the therapeutic management of AD with consideration of ethical concerns and international collaboration of interdisciplinary research to tackle the mechanisms of AD [14,15].

The limitations of current treatments can be countered by the use of complementary and alternative medicines. Highly nutritious culinary and medicinal mushrooms are rich sources of antioxidants that have been shown to delay the progression of neurodegenerative disorders [16,17]. Mushrooms contain a large number of bioactive compounds including alkaloids, flavonoids, polyketides, steroids, terpenes, polysaccharides, proteins, micronutrients, and unsaturated fatty acids. The phenolic compounds have the highest antioxidant activities and are mainly responsible for attenuating oxidative damage. *Hericium erinaceus* (HE), also known as the monkey’s head mushroom, lion’s mane mushroom, or Yamabushitake, is commonly found in East Asia [18] (Figure 1). It is well-known for its diverse therapeutic activities, including neuroprotection [16,17,19,20] and neuroregeneration [21,22,23,24], which are attributed to its neurogenesis, antioxidative, and anti-neuroinflammatory functions [17,19,20,23,25].

The structural characterization and isolation of compounds in the polysaccharide fraction (mainly beta-glucans) and secondary metabolites (e.g., hericenones, erinacines, hericerins, resorcinols, monoterpenes, diterpenes, steroids, and volatile aromatic compounds) of HE fruiting bodies and mycelia have been shown to have immune-modulatory activities, neurogenesis, antioxidative, anti-inflammatory, and anti-cancer properties [17,19,20,23]. These neurotrophic compounds have potential as treatments for AD, due to their potential to penetrate through the blood–brain barrier [26]. Additionally, recent studies have reported that HE can modulate neural activity, such as increasing synaptic plasticity, reducing apoptosis, decreasing Aβ plaques, and inhibiting acetylcholinesterase (AChE) and BACE1 [27], supporting its therapeutic potential in the management of AD. This narrative review discusses recent clinical and animal studies on the therapeutic efficacy of HE for AD.

## 2. Etiology of Alzheimer’s Disease

Alzheimer’s disease is associated with complex multifaceted pathologies, leading to neuronal dysfunction and degeneration. In AD, the deposition of senile plaques and intraneuronal NFTs tend to form in stereotypical neuroanatomical patterns throughout the brain [3,4,5]. The accumulation of Aβ peptides arises from the sequential cleavage of amyloid precursor protein (APP) by β-Site APP-cleaving enzyme 1 (BACE1) to generate soluble C99 and APPβ fragments, followed by further cleavage by γ-secretase [3,4,5]. It has also been shown that mutations in Presenilin-1 and -2 in AD can modify the activity of γ-secretase, which increases the accumulation of Aβ peptides [28]. The Aβ peptides form soluble oligomers that aggregate and deposit as senile plaques, suppressing the growth and differentiation of neuronal cells. The downregulation of β-catenin by oligomeric Aβ senile plaques consequently impairs Wnt/β-catenin signaling, which is an essential signaling pathway of neurogenesis [29]. Inhibition of this pathway increases the production of Aβ senile plaques and tau hyperphosphorylation through activation of GSK3β signaling [30]. Abnormally hyperphosphorylated tau form NFTs that also contribute to the suppression of neurogenesis [31]. The NFTs disrupt microtubule-associated proteins that regulate microtubule dynamics, which impairs axonal transport and leads to the progressive loss of neuronal cells [32,33].

Oxidative stress, which is the overproduction of reactive oxygen species (ROS), has been implicated in the progression of AD. ROS participates in various cellular functions, including gene transcription and signal transduction; however, overproduction of ROS and free radicals lead to the deterioration of cellular antioxidant defense mechanisms [34]. Intracellular ROS is observed to be generated by various endogenous and exogenous sources. For instance, mitochondria generate approximately 90% of cellular ROS. The mitochondrial electron transport chain (ETC) is considered to be the primary endogenous source of ROS including superoxide (O^2−^•), hydrogen peroxide (H_2_O_2_), singlet oxygen (^1^O_2_), nitric oxide (NO), and hydroxyl radical (OH•), which are involved in cellular functioning and signaling [35,36].

The univalent metabolic reduction status of oxygen can lead to the production of potentially harmful ROS [35]. Mutations and metabolite deficits can result in a major shift in the redox balance and mitochondrial dysfunction, which inhibit enzyme and protein synthesis, leading to excessive ROS production. In the brains of AD patients, increased oxygen consumption, elevated polyunsaturated lipid levels, and impaired antioxidant defense systems lead to further generation of ROS. Neuron and glial cells are vulnerable to oxidative stress and the accumulation of excess ROS modifies the function of major biomolecules, including nucleic acids (DNA, RNA), proteins, and lipids [35]. In addition, alterations in the activity of antioxidant enzymes (e.g., glutathione peroxidase, superoxide dismutase) have been shown to decrease the proliferation rate of neuronal cells in AD [37]. 

The accumulation of NFTs and Aβ deposition have demonstrated to activate neuroinflammation, which plays a harmful role in AD by causing the increased inflammatory mediators expression in the vicinity of Aβ peptide deposits and NFTs [27]. The neuroinflammation defense system is supported by glial cells such as microglia that act as the primary defense line in the CNS. Glial cells phagocytize exogenous and harmful substances and also produce inflammatory cytokines including interferon-gamma (IFN-γ), interleukin one beta (IL-1β), IL-6, and tumor necrosis factor-alpha (TNF-α) [27,38,39]. Chronic and sustained activation of microglia causes overproduction of neurotoxins and cytokines, resulting in neuroinflammation, triggering a series of intrinsic apoptotic pathways, and gradually leading to neuronal cell death [27,39]. Insult to the central nervous system also results in molecular, structural, and functional modifications to astrocytes, a process known as reactive gliosis [4]. Reactive gliosis has beneficial effects during the initial phases of AD development by eliminating neurotoxic peptides and inducing inflammatory signals [4,40]. However, sustained and excessive activation of pro-inflammatory mediators and ROS levels cause adverse neurotoxic effects on neighboring neurons [27]. Research has shown inflammatory mediators have effects in the pathogenesis of AD through the generation of Aβ peptides from amyloid precursor proteins via β- and γ-secretase enzymes [27,41]. 

Lastly, altered neurotransmitter expression and loss of neuronal synapses are closely associated with AD pathogenesis [5]. Acetylcholine (ACh), an essential neurotransmitter that contributes to learning and memory, is found to be deficient in AD [5,42,43]. A study by Slotkin et al. found that reduced choline acetyltransferase levels resulted in decreased synthesis of ACh. This further reduced its uptake by acetylcholine receptors (AChRs), leading to the loss of memory function [42,43]. The presence of Aβ peptides and NFTs in AD was also found to influence the expression of acetylcholinesterase, which further affects the synthesis of ACh [44]. Reduced levels of AChRs have also been observed in the mild condition of AD, leading to a decreased uptake of ACh [45]. Glutamate, an essential excitatory neurotransmitter associated with learning and memory, is also impaired in AD. Glutamate uptake or recycling can be impaired in AD due to alterations in the expression of glutamate transporters. It is well-established that increased presynaptic release of glutamate and failure of glutamate uptake lead to the activation of NMDA ionotropic receptors [46]. Excessive activation of NMDA receptors leads to excitotoxicity and decreases neuronal cell proliferation and survival [47]. Mitochondrial dysfunction is also known to be a contributing factor to glutamate excitotoxicity in AD [46].

## 3. Preclinical Studies of *Hericium erinaceus* in AD

*Hericium erinaceus* has recently attracted considerable research attention due to its therapeutic potential in treating the debilitating conditions in AD [48,49,50]. This review analyzes six essential animal studies on the therapeutic potential of HE in AD models (Table 1). The commonly used HE products include erinacine A-enriched HE-mycelia (HE-A), ethanol extracts of erinacine A-enriched HE-mycelia (HE-Et), and erinacine S-enriched HE mycelia (HE-S). Culinary and medicinal mushrooms are consumed by humans as a functional food and as a nutraceutical source. Hence, the use of mushrooms with beneficial health effects in humans requires in vivo experimentation to ensure their safety and efficacy. Aqueous extracts of HE rich in polysaccharides were shown to promote neuronal growth and differentiation [17,23,51,52,53] via the reduction in endoplasmic reticulum stress-induced cell atrophy, expression of neurotrophic factors in astrocytes, and decrease in neurodegenerative-induced cell atrophy [51,54,55,56]. The mechanism of action of HE has been further investigated in several preclinical studies. The effects of HE were investigated in Aβ (25–35)-treated mice that had learning and memory deficits, as seen by reduced discrimination in the novel object recognition (NOR) test and poor working memory in the Y-maze test [48]. Oral administration of HE rescued the learning and memory impairment in Aβ (25–35)-injected mice through elevation of nerve growth factor (NGF) mRNA abundance in the hippocampus, suggesting that it can enhance hippocampal neurogenesis. However, this study only investigated NGF expression and did not report the physiological effects of HE treatment on learning and memory. Moreover, the short duration of the HE treatment might not provide insights on its long-term beneficial effects. 

Tsai-Teng et al. (2016) further investigated the effects of HE-A and HE-Et on APPswe/PS1dE9 mice, which resulted in a reduction in Aβ buildup in the cerebral cortex in a short-term and long-term administration treatment paradigm [49]. After 30 days of oral administration, there were decreased astrocytes and plaque-activated microglia in the cerebral and hippocampal regions. Additionally, there was an increase in the ratio of NGF to neural growth factor precursor, indicating hippocampal neurogenesis, which correlated with the ameliorated nesting behavior of the AD mice. Long-term administration demonstrated improved nesting behavior, which has been previously associated with progressive cognitive impairment in APPswe/PS1dE9 mice, suggesting the potential role of HE as a treatment for AD [57].

Tzeng et al. (2018) also conducted a short-term and long-term administration treatment paradigm that reported the effects of ethanol extracts for HE-A and HE-S in APPswe/PS1dE9 mice [50]. After 1 month of oral administration, both HE-A and HE-S showed beneficial effects, including inhibiting cerebral plaque growth and reducing glial cell activation. Moreover, HE-A reduced insoluble Aβ fragments and the C-terminal fragment of APP, leading to hippocampal neurogenesis and attenuating Aβ production. After 100 days, the increase in neurogenesis was found to be correlated with recovering behavioral and memory deficits in the nesting, burrowing, and Morris water maze (MWM) tests.

Zhang et al. (2016) investigated the mechanisms of polysaccharide-enriched aqueous extract of HE mycelia in an AD model of mice which was generated by subcutaneously injecting D-galactose and intragastrically administrating aluminum chloride (AlCl_3_) [47]. The brains of AD mice demonstrated reduced AChE and acetyltransferase (ChAT) levels, resulting in learning and memory deficits [5]. Mice treated with the HE extracts depicted dose-dependent increased concentrations of AChE and ChAT in the hypothalamus and blood serum, which correlated with enhanced learning and memory in the behavioral tests, suggesting that the neuroprotective effects of polysaccharide-enriched HE is mediated through the cholinergic signaling pathway. 

A recent study by Cordaro et al. (2021) demonstrated the effects of HE in a Wistar rat model of AD induced by AlCl_3_, which exhibited AlCl_3_ accumulation in the hippocampus [58]. Treatment with HE ameliorated the AlCl_3_ accumulation, improved behavioral deficits, and enhanced hippocampal neurogenesis, leading to reduced levels of tau phosphorylation, APP overexpression, and Aβ accumulation. Furthermore, the HE treatment was found to mitigate oxidative stress through suppressing NLRP3 inflammasome activation. These results indicate that HE also has anti-oxidative and anti-inflammatory activity.

Lee et al. (2021) expanded the previous research on the effects of HE-A on brain aging, learning, and memory [50,59]. They found that a preparation of HE-A significantly improved cognitive function in active and passive avoidance behavioral tests and delayed cognitive degeneration due to aging. In addition, HE-A significantly decreased the levels of induced nitric oxidase synthase (iNOS), thiobarbituric acid-reactive substances (TBARS), and 8-hydroxy-2′-deoxyguanosine (8-OHdG) in a dose-dependent manner, leading to the attenuation of oxidative stress. The findings support that HE has beneficial anti-oxidative and anti-inflammatory properties.

**Table 1 cells-11-02284-t001:** Preclinical studies of *Hericium erinaceus* in AD animal models.

Authors	Animal Models	Treatment Method	Behavioral Test	Behavioral Outcome	Mechanism and Physiological Effect
Mori et al., 2011 [48]	5-week-old male ICR mice with Aβ (25–35) and Aβ (35–25)	10 μg of amyloid β (25–35) peptide administered intracerebroventricularly on days 7 and 14 and fed with HE diet (powdered mixture of normal diet and HE), containing 5.5% of (*w*/*w*) for 23 days	Y-Maze test NOR	No significant difference observed in alternation behavior between Aβ (25–35) and Aβ (35–25) group HE increased exploration time for novel object than for familiar object in Aβ (25–35) mice, but not Aβ (35–25) mice	Increased hippocampal NGF mRNA expression
Tsai-Teng et al., 2016 [49]	5-month-old female APPswe/PS1dE9 double transgenic mice	Short-term: Oral administration of HE-A and HE-Et (300 mg/kg/day) for 30 days Long-term: Oral administration of HE-My (300 mg/kg/day) for 70–90 days	Nesting	HE-My for 81 days improved nesting behaviors	HE-A or HE-Et for 30 days: Eliminated Aβ plaque burden Prevented recruitment and activation of plaque-associated microglia and astrocytes Promoted proliferation of neuron progenitors Increased neuronal proliferation in the dentate gyrus
Tzeng et al., 2018 [50]	5-month-old female APPswe/PS1dE9 double transgenic mice	Short-term: HE-A or HE-S (30 mg/kg/day) administered through gavage with vehicle for 30 days Long-term: Oral administration of HE-A (10 and 30 mg/kg/day) for 100 days	Burrowing Nesting MWM	HE-A ameliorated learning and spatial memory during the probe trial Deficits in spontaneous burrowing behavior significantly recovered at both 10 and 30 mg/kg of HE-A Impaired nesting behavior significantly recovered at 30 mg/kg of HE-A	HE-A and HE-S decreased Aβ plaque burden and increased cerebral Aβ degradation HE-A decreased Aβ accumulation by inhibiting Aβ production in the cerebrum HE-A and HE-S reduced activation of glial cells in the cerebrum HE-A and HE-S promoted neurogenesis and dendritic complexity in the hippocampus
Zhang et al., 2016 [47]	10-week-old Balb/c female mice with 120 mg/kg of D-gal 20 mg/kg of kg of AlCl_3_	Subcutaneous injection of 120 mg/kg of D-gal and intragastric administration of 20 mg/kg of AlCl3 once per day for 10 weeks Intragastric administration of polysaccharide-enriched aqueous extract of HE mycelia at dose of 0.3, 1.0, and 3.0 g/kg for 4 weeks	Autonomic activities test MWM Fatigue rotarod test	HE enhanced vertical and horizontal movements in the autonomic activity test HE ameliorated rotarod test endurance time HE reduced MWM escape latency time	Increasing the dose of HE-enhanced AChE and ChAT concentrations in the serum and hypothalamus
Cordaro et al., 2021 [58]	6–8-week-old male Wistar rats with AlCl_3_	Intraperitoneally administered 70 mg/kg of AlCl3 daily for 6 weeks Control + HE: Oral administration of HE (200 mg/kg) daily by gavage AD + HE: Oral administration of HE (200 mg/kg) daily by gavage	MWM EPM NOR	HE increased animal permanence in target quadrant HE increased time of novel object recognition with high discrimination ratio	HE reduced AlCl_3_-induced CA1 neuronal degeneration HE increased Nrf2 expression in the hippocampus HE increased antioxidant defense including SOD, CAT, and GSH levels HE reduced NLRP3 inflammasome activation HE decreased phosphorylated Tau, APP overexpression, and Aβ aggregation
Lee et al., 2021 [59]	3-month-old male and female (SAMP8) mice	Low-dose group (108 mg/kg/bw/day), intermediate-dose group (215 mg/kg/bw/day), and high-dose group (431 mg/kg/bw/day) of oral HE-A administration for 13 weeks	Passive Avoidance Task Active Shuttle Avoidance Task	HE-A significantly increased number of avoidance responses Latency time after training increased for passive avoidance test in HE-A groups	HE-A lowered iNOS expression, lowering oxidative stress/inflammation HE-A decreased TBARS levels, decreasing lipid peroxidation HE-A resulted in a downward trend in Aβ plaque (%)

Abbreviations: HE, *Hericium erinaceus*; APPswe, Amyloid precursor protein; PS1dE9, Presenilin-1; BrdU, Bromodeoxyuridine; HE-A, erinacine A-enriched *Hericium erinaceus* mycelia; ADL, Activities of daily living; HE-Et, ethanol extract of erinacine A-enriched *Hericium erinaceus* mycelium; HE-S, ethanol extract of erinacine S-enriched *Hericium erinaceus* mycelium; MWM, Morris water maze; Aβ, Amyloid-beta; Balb/c, Bagg and albino; AlCl_3_, Aluminum; AChE, Acetylcholinesterase; ChAT, Choline acetyltransferase; EPM, Elevated plus maze; NOR, Novel objection recognition; CA1, Carbonic anhydrase 1; Nrf2, Nuclear factor-erythroid 2-related factor 2; SOD, Superoxide dismutase; CAT, Catalase; GSH, Glutathione; NLRP3, NLR family pyrin domain containing 3; NGF, Nerve growth factor; mRNA, Messenger RNA; SAMP8, Senescence accelerated mouse prone 8; iNOS, Induced nitric oxidase synthase; TBARS, Thiobarbituric acid-reactive substances.

## 4. Clinical Studies of *Hericium erinaceus* in AD

Only three clinical trials have thus far investigated the potential therapeutic effects of HE on AD in humans (Table 2). In a completed clinical trial conducted by Li et al. (2020), the safety and efficacy of HE-A were investigated in patients with a mild form of AD [60]. Participants were administered daily for 49 weeks with 350 mg mycelia-based capsules of erinacine A. The treatment group depicted a notable improvement in the activity of daily living (ADL), cognitive ability, and mini-mental state scores. Moreover, HE-A treatment improved contrast sensitivity in the ophthalmologic examination compared to the placebo group. Although HE-A was largely found to be safe and effective, four participants dropped out of the study due to adverse reactions, including nausea, abdominal pain, and skin rash. Clinical studies with a much larger sample size are required to further verify the neurocognitive benefits of HE-A in AD patients.

A double-blind, parallel group study by Mori et al. (2009) investigated the effects of oral administration of four 250 mg capsules containing 96% HE three times daily for 16 weeks in Japanese women and men with mild cognitive impairment. The treatment group demonstrated significant improvement in cognitive functioning scores compared to the placebo group with no apparent side effects [61]. Although there was a positive correlation between the treatment and improved cognitive function, this only lasted during the drug administration period and cognitive function scores decreased thereafter, which suggests the need for long-term HE use for the management of AD. 

The most recent study was a double-blind, parallel group clinical trial by Saitsu et al. examining the effects of the consumption of four HE supplements containing 0.8 g of powdered fruiting bodies daily for 12 weeks. Cognitive abilities were assessed by the Mini-Mental State Examination (MMSE), standard verbal paired-associate learning, and Benton visual retention tests [62]. The HE treatment was found to prevent short-term memory decline and improve cognitive function in the MMSE, indicating the beneficial effects on neural network regeneration and its overall safety.

## 5. Mechanisms of *Hericium erinaceus* in AD

Several erinacines and hericenones have been isolated from the fruiting bodies and mycelia of HE, respectively [63]. Among them, 15 erinacines and cyathane diterpenoids were reported to possess various biological activities. Erinacines A–I were demonstrated to have neuroprotective properties through enhancing the release of neurotrophic factors, increasing the expression of insulin-degrading enzymes (erinacines A and S), reducing Aβ aggregation, and managing neuropathic pain (erinacine E) (Figure 2) [63]. The majority of hericenones were demonstrated to have been correlated with improved cognitive function through the activation of NGF synthesis in astrocytes, whereas erinacine B was found to prevent thrombosis, increase cerebral blood flow, and confer protection against cerebrovascular risk and vascular dementia [62,64]. 

### 5.1. Anti-Amyloidogenic Functions

*Hericium erinaceus* was reported to have anti-amyloid properties in reducing Aβ synthesis and accumulation and protecting neuronal cells against Aβ cytotoxicity [28]. Multiple mechanisms have been implicated in the clearance of Aβ plaque, including a reduction in CTF-β, SDS-soluble Aβ1-40, and SDS-insoluble Aβ levels [50]. Treatment with EAHEM reduced levels of Aβ1-42, which is the variant most prone to aggregation. Moreover, HE was found to prevent the deposition of Aβ peptides through the proteolytic degradation of Aβ and APP intracellular domain (AICD) by insulin-degrading enzyme (IDE) [65]. Farris et al. (2004) reported that IDE is a key proteolytic enzyme in Aβ reduction. Rats with partial loss-of-function mutation of IDE and an IDE-knockout mouse model demonstrated enhanced Aβ accumulation in the cerebral region [66]. In addition, a mouse model with AD-associated ApoE4 allele showed reduced levels of IDE associated with increased Aβ [67]. Remarkably, Tzeng et al. (2018) found that both HE-A and HE-S were able to increase the expression of IDE in AD animal models accompanied by a reduction in Aβ, as seen in the immunohistochemical analysis [50].

### 5.2. Anti-Oxidative Function

Several studies have suggested that the neuroprotective effects of HE result from upregulated antioxidant enzymes (e.g., glutathione peroxidase, catalase, and SOD) and reduced MDA levels that are implicated in the cellular defense mechanisms against ROS [68]. Furthermore, HE was shown to exert its antioxidant effects through the regulation of the transcriptional activity of nuclear factor-erythroid 2-related factor 2 (Nrf2) [69]. The Nrf2 signaling pathway regulates genes encoding various proteins that function as endogenous stress–response proteins, antioxidant enzymes, and redox-maintaining factors [58]. 

The antioxidant capacity of HE has been demonstrated in several preclinical animal studies. In a study by Lee et al. (2021), administration of increasing concentrations of EAHEM in SAMP8 mice over 13 weeks restored the level of TBARS, which is an index of lipid peroxidation [59]. This restoration is important considering that the long-term accumulation of lipid peroxidation is a key contributor to the aging brain and cognitive deterioration [70]. Besides, an ethanol extract of HE was also found to reduce apoptotic activity by inhibiting Bax/Bcl-2 and caspase-3 signaling pathways in a cellular model of glutamate-induced oxidative stress [71].

### 5.3. Anti-Neuroinflammation

Recently, Cordaro et al. (2021) demonstrated the anti-neuroinflammatory effects of HE by ameliorating NLRP3 inflammasome activation, which was found to involve the antioxidant properties of HE [58]. The inflammasome complex consists of various proteins, including DAMPS or PAMPS receptor (damage- or pathogen-derived molecular patterns), NLRP3 (NLR family pyrin domain containing 3), and pro-caspase-1 activated through ASC (apoptosis-associated speck-like protein containing a caspase recruitment domain) [58,72]. The NLRP3 inflammasome can sense a wide range of stimuli to trigger inflammation, mediating the activation of DAMPs or PAMPs, recruitment of ASC, and cleavage of pro-caspase-1 (pro-IL1β and pro-IL18) to generate pro-inflammatory cytokines [58,73]. The findings by Cordaro et al. (2021) revealed the anti-inflammatory mechanisms through the downregulation of the inflammasome network by decreasing the expression levels of ASC, NLRP3, and pro-caspase-1. Additionally, HE was also shown to inhibit the activation of NF-kB, a pro-inflammatory transcription factor [58].

In addition, HE was found to reduce the inflammatory responses by regulating iNOS expression. Three nitric oxide synthase (NOS) isoforms (i.e., neuronal NOS (nNOS), endothelial NOS (eNOS), and inducible NOS (iNOS)) [59], and among these isoforms, increased iNOS expression has been correlated with oxidative stress and inflammatory processes [74]. Lee et al. (2021) observed a reduction in iNOS expression in mice administered EAHEM, which suggests that its neuroprotective effects were mediated through the attenuation of inflammation and oxidative stress. 

### 5.4. Neurotrophic Mechanisms

*Hericium erinaceus* has been shown to stimulate the release of neurotrophic factors, including NGF and brain-derived neurotrophic factors, which are known to regulate the development, maintenance, function, and survival of neuronal cells [54]. Apart from being the major players in neuroplasticity, these neurotrophic factors can also activate neurogenesis and protect neuronal cells against apoptosis. HE extracts stimulate NGF release by promoting NGF mRNA expression in astrocytes via the c-jun N-terminal kinase signaling [75]. The increased levels of NGF released from astrocytes transmit into the nerve cells and have been associated with neurogenesis and neuroplasticity in the hippocampus, pituitary glands, and cerebral cortex [16,76]. The binding of NGF to tropomyosin receptor kinase A (TrkA) receptors results in the activation of extracellular signal-regulated protein kinase (Erk)-cyclic adenosine monophosphate (cAMP)-response element-binding protein (CREB) signaling cascade, which modulates proliferation, maintenance, and memory development in neural precursor cells [76]. Furthermore, NGF-mediated neuronal differentiation also promoted an extensive mitochondrial remodeling [77] and increased fusion proteins (Mfn2 and Opa1), Drp1-dependent mitochondrial fission, activation of Sirt3 and PPARγ, and mtTFA transcription factors, ultimately controlling bioenergetic capacity. Martorana et al. (2018) reported that NGF was important for mitochondrial remodeling and contributed to neurogenesis and nerve regeneration [77]. 

Various studies have shown that HE treatments can have long-lasting effects on increasing Ki67-positive, PCNA-positive, and BrdU immunoreactive cells in the dentate gyrus of the hippocampus, leading to the development of neural progenitor cells in the hippocampus [16,78]. Current evidence suggests that the regulation of hippocampal neurogenesis by HE involves NGF by increasing its mRNA and protein expression levels, which also demonstrates the ability of HE bioactive compounds to pass through the blood–brain barrier [16,78].

### 5.5. Neurotransmission

The mechanism by which HE modulates the expression of neurotransmitters has been investigated in preclinical studies. Treatment with HE was found to improve cholinergic function by enhancing ACh and choline acetyltransferase levels in AD mouse models [47]. Brandalise et al. (2017) found that dietary HE supplementation enhanced the release of glutamate neurotransmitter from the hippocampal mossy fiber terminals, as evident by the increased spontaneous excitatory activities in the mossy fiber-CA3 synapses that were found to be dependent on glutamate release [79]. Further studies are required to examine the effects of HE on other memory-related neurotransmitters to better understand its modulating pathways.

Overall, the results of these studies indicate that HE treatments improved memory, which was accompanied with enhanced hippocampal neurogenesis and modulation of the anti-amyloidogenic, anti-oxidative, anti-neuroinflammatory, and neurotransmitter pathways (Figure 3).

## 6. Limitations and Future Perspective

*Hericium erinaceus* and its bioactive compounds have been shown to target neuropathological hallmarks associated with AD and has potential for treating the debilitating symptoms. Nevertheless, the precise mechanisms and therapeutic outcomes of HE in AD remain unclear and will require further research. For example, whether these compounds can cross the blood–brain barrier will require the investigation of their physicochemical and pharmacokinetic properties [16]. 

Considering only erinacines A and S have been revealed to possess beneficial effects in AD, there needs to be more research on other erinacines and cyathane diterpenoids to facilitate the development of new therapeutics. For this reason, bioreactor designs and scale-up principles could be ideal technologies to produce pure mycelial materials without any feedstock residue. Such processes would ensure that high-quality HE mycelia can be consistently produced by controlling the substrate, HE strain, growth conditions, and post-processing.

On the contrary, as a complementary and alternative medicine, HE may elicit its neuroprotective effects synergistically through multiple compounds and via multiple targets. Many herbal formulations and alternative medicines act synergistically through the various constituents to elicit their therapeutic effects [80,81]. However, the safety and effectiveness of some HE formulations have been questioned, mainly due to concerns about the proprietary formulas from different manufacturers [82,83]. Therefore, guidelines on good pharmacovigilance practices and monitoring the safety of HE formulations, as well as ethical concerns on memory modulation, are required to achieve quality compliance in clinical trials [14,15].

Recent progress in nanotechnology [84] and neuromodulation techniques [85,86,87,88,89] could pave the way for revolutionizing the development of compounds to tackle dementia. In particular, one could combine HE treatment with an invasive/non-invasive brain stimulation approach to enhance the therapeutic potential of memory function in AD patients [85,86,87,88,89]. In this respect, the development of nanotherapeutics with multi-functionalities has considerable potential to bridge the gap between the challenges associated with current therapeutics and their clinical application as treatments for AD. Biosynthesis of HE nanoparticles can enhance the delivery or transportation of bioactive compounds by preventing drug resistance, through increasing their bioavailability at the target sites and bioactivity that results in prolonged action of sustained drug release.

The therapeutic effects of HE have mostly been tested in preclinical animal models with only a few clinical studies. To overcome such limitations, different preparations and formulations of HE should be incorporated into phase 1 clinical trials to evaluate their safety in healthy volunteers. Presently, although medicinal mushrooms are readily available, information regarding their dosage, preparation, and manufacturing processes may differ significantly among manufacturers. The lack of standardized parameters in terms of dosage, adverse effects, and active ingredients may hamper conducting clinical studies and affect the reliability and validity of the outcomes [90]. Furthermore, a universal regulatory system needs to be established to ensure the safety and efficacy of phytopharmaceuticals, as evidence-based verification of the activity of alternative medicines including medicinal mushrooms is often not available.

## 7. Conclusions

Several preclinical and clinical investigations have established the therapeutic potential of HE in ameliorating behavior and cognitive function deficits, and overall beneficial effect in delaying AD pathogenesis. However, present research on the effects of HE, particularly their mechanisms of action, is still in the early phase. Further studies are required to establish the efficacy and safety of these compounds and to understand their specific mechanisms with interdisciplinary research collaboration of international societies [14,15].

## Figures and Tables

**Figure 1 cells-11-02284-f001:**
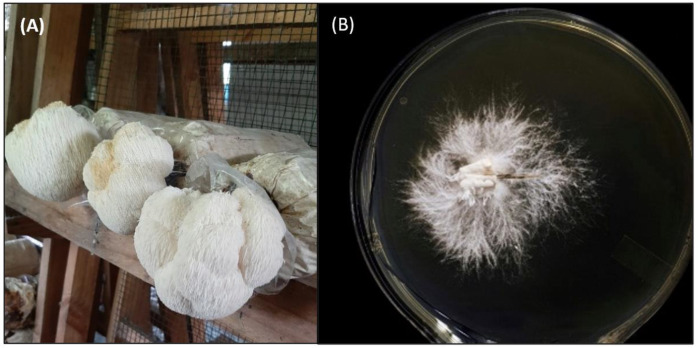
*Hericium erinaceus* fruiting body (**A**) and mycelia (**B**).

**Figure 2 cells-11-02284-f002:**
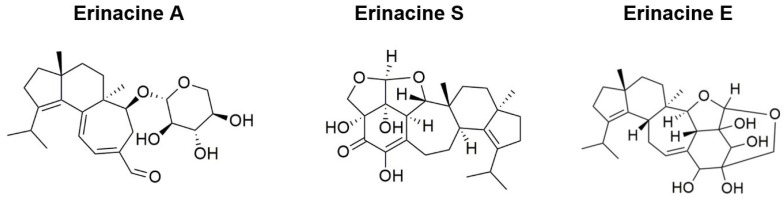
Bioactive compounds isolated from *Hericium erinaceus* with therapeutic effects on Alzheimer’s disease.

**Figure 3 cells-11-02284-f003:**
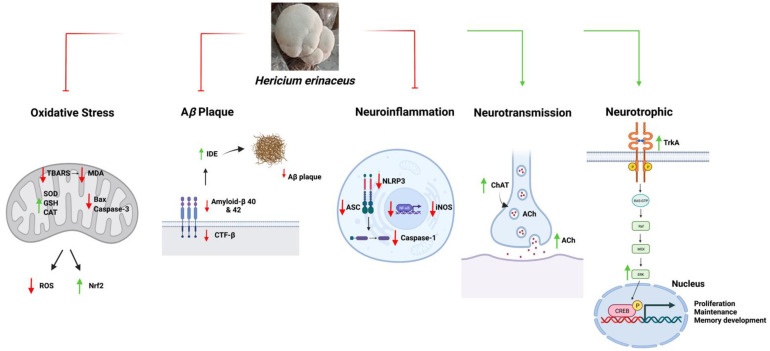
A schematic diagram summarizing the functions of *Hericium erinaceus* in AD. Abbreviations: ROS, Reactive oxygen species; BAX, Bcl-2- associated X protein; TBARS, Thiobarbituric acid reactive substances; MDA, Malondialdehyde; Nrf2, Nuclear factor-erythroid factor 2-related factor 2; SOD, Superoxide dismutase; CAT, Catalase; GSH, Glutathione; Bcl-2, B-cell lymphoma 2; SDS, Sodium dodecyl sulfate; CTF-β, Beta-carboxyl-terminal fragment; Aβ, Amyloid-beta; IDE, Insulin-degrading enzyme; iNOS, Nitric oxide synthase; ASC, Apoptosis-associated speck-like protein containing a caspase recruitment domain; NLRP3, NLR family pyrin domain containing 3; NF-kB, Nuclear factor-kappa B; ACh, Acetylcholine; ChAT, Choline acetyltransferase; TrkA, Tropomyosin receptor kinase A; RAS-GTP, Ras protein guanine triphosphatase; Raf, Rapidly accelerated fibrosarcoma; MEK, Mitogen-activated protein kinase; Erk, Extracellular signal-regulated kinase; CREB, cAMP-response element binding protein.

**Table 2 cells-11-02284-t002:** Clinical studies of *Hericium erinaceus* in AD individuals.

Authors	Disease, Clinical Phase, and Duration	Research Design	Administration Method	Dosage	Outcome Measures	Clinical Evaluation	Adverse Effects
Li et al., 2020 [60]	Mild Alzheimer’s disease Completed 52 weeks	Double-blind, two parallel groups, randomized, placebo-controlled 49 participants, 50 to 90 years old	Oral route	350 mg mycelia-based capsule containing 5 mg/g of erinacine A per day or identically appearing placebo capsules	Significant improvement in MMSE, IADL, CASI scores, and better contrast sensitivity than placebo HE-A was well-tolerated, safe, and improved cognitive function	Adverse events evaluation NPI MMSE CASI IADL Ophthalmic examination Biomarker collection Neuroimaging	1 subject lost to follow-up 3 subjects lost to unsatisfactory efficacy 4 subjects lost to side effects: nausea, abdominal discomfort, nausea, and skin rash
Mori et al., 2009 [61]	Mild cognitive impairment Completed 22 weeks	Double-blind, parallel group, placebo-controlled trial, randomized 30 participants, 50 to 80 years old	Oral route	Four 250 mg tablets of dry powder of HE (96% of HE) or three placebo tablets per day for 16 weeks	Improved cognitive function scale scores	HDS-R Adverse effect evaluation Blood chemistry	N.A.
Saitsu et al., 2019 [62]	Normal cognitive functions Completed 12 weeks	Double-blind, placebo-controlled trial, parallel group, randomized 34 participants, 50 years old and above	Oral route	Four HE supplements containing 0.8 g of powdered fruiting body or four placebo supplements per day for 12 weeks	Significantly improved cognitive functions in MMSE	MMSE Benton visual retention test S-PA	N.A.

Abbreviations: MMSE, Mini-mental state examination; IADL, Instrumental activities of daily living; CASI, Cognitive abilities screening instrument; HE-A, erinacine A-enriched *Hericium erinaceus* mycelia; NPI, Neuropsychiatric inventory; HDS-R, Revised Hasegawa Dementia scale; S-PA, Standard verbal paired-associate learning test.

## Data Availability

Not applicable.

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
