# Peer review of "The Monkey Head Mushroom and Memory Enhancement in Alzheimer’s Disease"

_cells, 2022, doi:10.3390/cells11152284_

Round 1

Reviewer 1 Report

This is a reasonably clear review on the potential use of the monkey head mushroom, Hericium erinaceus, for the treatment of Alzheimer’s disease (AD). Since there are still no good treatments for AD and some of the recently approved treatments have poor efficacy and are very expensive, further exploration of the use of natural products for disease treatment is warranted. However, there are a number of points, listed below, that need clarification or additional information in order to make this review useful to readers.

1. The authors need to provide a list of the different H. erinaceus products and their abbreviations that have been used for pre-clinical and clinical studies. There seem to be a variety of these and it is hard to keep track of them while reading the review.

2. lines 106-109: This sentence is not clear.

3. line 1124: Glutathione is not an enzyme. Please clarify what you mean here.

4. line 133: Is IL16 correct? Are you sure you don’t mean IL6?

5. lines 134-137: Neuroinflammation is not a pivotal function of microglia. It can result from microglial activation but it is not their function. Please correct.

6. lines 189-191: How do the authors know that hippocampal neurogenesis was responsible for the improvement in nesting behavior and what is the relevance of nesting behavior to AD?

7. lines 198-200: How do the authors know that neurogenesis was responsible for these improvements in behavior? Was neurogenesis inhibited and did that block the behavioral improvements? If not, then it is just a correlation and the authors need to make that clear.

8. line 204: What is the relevance of AlCl3 to AD?

9. lines 206-209: How do the authors know that the increased AChE and ChAT were responsible for the improvements in behavior? Did inhibition of AChE block the effects of the HE extract?

10. line 219: What model was used here?

11. Table 1: Mori et al-when was HE given with respect to Abeta injection?; Tsai-Teng et al.-why are there two different treatment paradigms? Explain better; Zheng et al. and Cordaro et al. – when was HE given relative to the toxins?; Lee et al-SAMP8 mice don’t have Abeta plaques so please clarify what is meant here.

12. Table 2: The authors should indicate the number of subjects in each trial. Li et al.-the adverse effects seem to contradict the outcome measures. Please clarify. Saitsu et al.-the authors should indicate that these subjects were cognitively normal.

13. lines 273-274: How was it demonstrated that the hericenones improved cognitive function by activating NGF synthesis in astrocytes. Was NGF synthesis specifically inhibited in astrocytes?

14. line 300: glutathione is not an enzyme. Please correct.

15. line 304: “endogenic” is not correct here. It should be “endogenous”.

16. lines 309-311: This statement needs a reference.

17. section 5.4: How does H. erinaceus stimulate NGF release from cells? How does it act specifically on astrocytes?

18. lines 368-370: This sentence makes no sense. Glutamate is a neurotransmitter but it is also a small molecule so its expression is not regulated.

19, lines 374-376: Why do the authors think that the anti-amyloidgenic, anti-oxidative, anti-neuroinflammatory or neurotransmitter pathways increase neurogenesis. What is the evidence for a direct cause and effect relationship? This is also a problem with the figure. None of the described references appear to discuss a direct relationship. There only appear to be correlations. Also, what is the direct evidence that neurogenesis improves memory?

20. lines 396-402: While this is interesting it needs to be put into context. The authors need to describe all of the components of HE, which ones have been shown to be active and how they are prepared and used for pre-clinical and clinical studies in a separate section near the beginning of the review.

Author Response

Response to Reviewer #1

[General Comments] This is a reasonably clear review on the potential use of the monkey head mushroom, Hericium erinaceus, for the treatment of Alzheimer’s disease (AD). Since there are still no good treatments for AD and some of the recently approved treatments have poor efficacy and are very expensive, further exploration of the use of natural products for disease treatment is warranted. However, there are a number of points, listed below, that need clarification or additional information in order to make this review useful to readers.

Response: Thank you very much for your comments that helped us improve this manuscript. We have read your comments carefully and taken them fully into account in the revision.

[Comment 1] The authors need to provide a list of the different H. erinaceus products and their abbreviations that have been used for pre-clinical and clinical studies. There seem to be a variety of these and it is hard to keep track of them while reading the review.

Response: Thank you for the suggestion. We have added a sentence of the commonly used HE products and their abbreviations:

The commonly used HE products include erinacine A-enriched HE-mycelia (HE-A), ethanol extracts of erinacine A-enriched HE-mycelia (HE-Et), and erinacine S-enriched HE mycelia (HE-S).” (p.4, line 170-172).

[Comment 2] lines 106-109: This sentence is not clear.

Response: Thank you for the comment. The sentence has been revise as follow:

Oxidative stress, which is the overproduction of reactive oxygen species (ROS), has been implicated in the progression of AD. ROS participates in various cellular functions, including gene transcription and signal transduction; however, overproduction of ROS and free radicals lead to the deterioration of cellular antioxidant defense mechanisms [1].” (p.3, line 108 – 112)

[Comment 3] line 124: Glutathione is not an enzyme. Please clarify what you mean here.

Response: Thank you for pointing out the mistake. It should be glutathione peroxidase. (p.4, line 127)

[Comment 4] line 133: Is IL16 correct? Are you sure you don’t mean IL6?

Response: Thank you for correcting us. We mean IL-6. (p.4, line 137).  

[Comment 5] lines 134-137: Neuroinflammation is not a pivotal function of microglia. It can result from microglial activation but it is not their function. Please correct.

Response: Thank you for the comment, we have removed that neuroinflammation is a pivotal function and added that it results from microglial activation:

“Chronic and sustained activation of microglia causes overproduction of neurotoxins and cytokines, resulting in neuroinflammation, triggering a series of intrinsic apoptotic pathways, and gradually leading to neuronal cell death [2, 3].” (p.4, line 138 - 140)

[Comment 6] lines 189-191: How do the authors know that hippocampal neurogenesis was responsible for the improvement in nesting behavior and what is the relevance of nesting behavior to AD?

Response: Thank you for the comment, as the paper did not specify if there is a direct cause, we have put that there is a correlation and added the relevance of nesting behavior to AD:

“Additionally, there was an increase in the ratio of NGF to neural growth factor precursor, indicating hippocampal neurogenesis, which correlated with the ameliorated nesting behavior of the AD mice. Long-term administration demonstrated improved nesting behavior which has been previously associated with progressive cognitive impairment in APPswe/PS1dE9 mice, suggesting the potential role of HE as a treatment for AD [4].” (p.5, line 195 - 200)

[Comment 7] lines 198-200: How do the authors know that neurogenesis was responsible for these improvements in behavior? Was neurogenesis inhibited and did that block the behavioral improvements? If not, then it is just a correlation and the authors need to make that clear.

Response: Thank you for the comment, as the paper did not indicate a direct association through blocking neurogenesis to see if the behavior was affected, we have clarified that a correlation was observed:

“Moreover, HE-A reduced insoluble Aβ fragments and the C-terminal fragment of APP, leading to hippocampal neurogenesis and attenuating Aβ production. After 100 days, the increase in neurogenesis was found correlated with recovering behavioral and memory deficits in the nesting, burrowing, and Morris water maze (MWM) tests.” (p.5, line 206 - 210)

[Comment 8] line 204: What is the relevance of AlCl3 to AD?

Response: Thank you for the question. The paper generated an AD model of mice by combining AlCl3 with d-galactose, and this sentence has been rephrased to make it clearer:

“Zhang et al. (2016) investigated the mechanisms of polysaccharide-enriched aqueous extract of HE mycelia in an AD model of mice which was generated by subcutaneously injecting d-galactose and intragastrically administrating aluminum chloride (AlCl3) [5].”  (p.5, line 212 - 215)

[Comment 9] lines 206-209: How do the authors know that the increased AChE and ChAT were responsible for the improvements in behavior? Did inhibition of AChE block the effects of the HE extract?

Response: Thank you for the comment, as the paper did not show if inhibition of AChE block affects the HE extract, we have clarified that a correlation was observed:

“Mice treated with the HE extracts depicted dose-dependent increased concentrations of AChE and ChAT in the hypothalamus and blood serum, which correlated with enhanced learning and memory in the behavioral tests” (p.5, line 216-218)

[Comment 10] line 219: What model was used here?

Response: Thank you, we have added which model was used:

“Wistar rat model of AD induced by AlCl3 (p.5, line 222 - 223)

[Comment 11] Table 1: Mori et al-when was HE given with respect to Abeta injection?; Tsai-Teng et al.-why are there two different treatment paradigms? Explain better; Zheng et al. and Cordaro et al. – when was HE given relative to the toxins?; Lee et al-SAMP8 mice don’t have Abeta plaques so please clarify what is meant here.

Response: Thank you for the comments, we have added the additional information in the table for Mori et al., Tsai-Teng et al. (also further explained in the preclinical section (p.5),  Zheng et al., and Cordaro et al. For Lee et al. the study mentioned and depicted the results of the amyloid β plaques in the SAMP8 male and female mice. (Table 1)

[Comment 12] Table 2: The authors should indicate the number of subjects in each trial. Li et al.-the adverse effects seem to contradict the outcome measures. Please clarify. Saitsu et al.-the authors should indicate that these subjects were cognitively normal.

Response: Thank you for the suggestion, we have indicated the number of subjects in each trial. For Li et al., the paper mentioned that while there were adverse effects noted, overall, the treatment was safe and well-tolerated. For Saitsu et al., normal cognitive functions have been added to the table. (Table 2)

[Comment 13] lines 273-274: How was it demonstrated that the hericenones improved cognitive function by activating NGF synthesis in astrocytes. Was NGF synthesis specifically inhibited in astrocytes?

Response: Thank you for the comment, as the paper did not show if inhibition of NGF synthesis in astrocytes improved cognitive function, we have clarified that a correlation has been found:

“The majority of hericenones were demonstrated to have been correlated with improved cognitive function through the activation of NGF synthesis in astrocytes” (p.13, line 284 - 285)

[Comment 14] line 300: glutathione is not an enzyme. Please correct.

Response: Thank you for pointing out the mistake. It should be glutathione peroxidase. (p.13, line 311)

[Comment 15] line 304: “endogenic” is not correct here. It should be “endogenous”.

Response: Thank you for pointing out the error, it has been fixed. (p.13, line 316)

[Comment 16] lines 309-311: This statement needs a reference.

Response: Thank you for the comment, we have added a reference. (p. 14, line 323)

[Comment 17] section 5.4: How does H. erinaceus stimulate NGF release from cells? How does it act specifically on astrocytes?

Response: Thank you for the question, we have added more clarity regarding the NGF release’s signaling pathway from cells:

“HE extracts stimulate NGF release by promoting NGF mRNA expression in astrocytes via the c-jun N-terminal kinase signaling [6]. The increased levels of NGF released from astrocytes transmit into the nerve cells and have been associated with neurogenesis and neuroplasticity in the hippocampus, pituitary glands, and cerebral cortex [7, 8].” (p.14, line 355 - 359)

[Comment 18] lines 368-370: This sentence makes no sense. Glutamate is a neurotransmitter but it is also a small molecule so its expression is not regulated.

Response: Thank you for the comment. We have revised this part as follows:

“Brandalise et al. (2017) found that dietary HE supplementation enhanced the release of glutamate neurotransmitter from the hippocampal mossy fiber terminals, as evident by the increased spontaneous excitatory activities in the mossy fiber-CA3 synapses that were found to be dependent on glutamate release [9].” (p. 15, line 383 - 386)

[Comment 19] lines 374-376: Why do the authors think that the anti-amyloidgenic, anti-oxidative, anti-neuroinflammatory or neurotransmitter pathways increase neurogenesis. What is the evidence for a direct cause and effect relationship? This is also a problem with the figure. None of the described references appear to discuss a direct relationship. There only appear to be correlations. Also, what is the direct evidence that neurogenesis improves memory?

Response: Thank you for the comment, we have updated the figure as no direct cause and effect relationship has been found in the research studies analyzed. Neurogenesis in these studies occurs in the hippocampus that plays a role in learning and memory, which is correlated with improved memory:

“Overall, the results of these studies indicate that HE treatments improved memory which was accompanied with enhanced hippocampal neurogenesis and modulation of the anti-amyloidogenic, anti-oxidative, anti-neuroinflammatory, and neurotransmitter pathways (Figure 3).” (p.15, line 390 - 393) (Figure 3)

[Comment 20] lines 396-402: While this is interesting it needs to be put into context. The authors need to describe all of the components of HE, which ones have been shown to be active and how they are prepared and used for pre-clinical and clinical studies in a separate section near the beginning of the review.

Response: Thank you for the comment and suggestion. Under the mechanisms section, we have mentioned the common and active components of HE, two of which have been discussed in detail in the pre-clinical section. (p.13, line 278 - 287)

References:

[1]        Poddar MK, Apala C, Soumyabrata B. Neurodegeneration: Diagnosis, Prevention, and Therapy. 2021.

[2]        Kushairi N, Tarmizi NAKA, Phan CW, Macreadie I, Sabaratnam V, Naidu M, et al. Modulation of neuroinflammatory pathways by medicinal mushrooms, with particular relevance to Alzheimer's disease. Trends in Food Science & Technology 2020;104:153-62.

[3]        Harry GJ, Kraft AD. Neuroinflammation and microglia: considerations and approaches for neurotoxicity assessment. Expert Opin Drug Metab Toxicol 2008;4(10):1265-77.

[4]        Filali M, Lalonde R. Age-related cognitive decline and nesting behavior in an APPswe/PS1 bigenic model of Alzheimer's disease. Brain research 2009;1292:93-9.

[5]        Zhang J, An S, Hu W, Teng M, Wang X, Qu Y, et al. The Neuroprotective Properties of Hericium erinaceus in Glutamate-Damaged Differentiated PC12 Cells and an Alzheimer’s Disease Mouse Model. International Journal of Molecular Sciences 2016;17(11):1810.

[6]        Mori K, Obara Y, Hirota M, Azumi Y, Kinugasa S, Inatomi S, et al. Nerve Growth Factor-Inducing Activity of Hericium erinaceus in 1321N1 Human Astrocytoma Cells. Biological and Pharmaceutical Bulletin 2008;31(9):1727-32.

[7]        Ji S, Wu H, Ding X, Chen Q, Jin X, Yu J, et al. Increased hippocampal TrkA expression ameliorates cranial radiation‑induced neurogenesis impairment and cognitive deficit via PI3K/AKT signaling. Oncol Rep 2020;44(6):2527-36.

[8]        Chong PS, Fung M-L, Wong KH, Lim LW. Therapeutic Potential of Hericium erinaceus for Depressive Disorder. International Journal of Molecular Sciences 2020;21(1).

[9]        Brandalise F, Cesaroni V, Gregori A, Repetti M, Romano C, Orrù G, et al. Dietary Supplementation of<i> Hericium erinaceus</i> Increases Mossy Fiber-CA3 Hippocampal Neurotransmission and Recognition Memory in Wild-Type Mice. Evidence-Based Complementary and Alternative Medicine 2017;2017:3864340.

Reviewer 2 Report

 In this review Yanshree et colleagues report a wide view for the therapeutical use of  monkey head mushroom as memory enhancer on AD. The review is concise yet complete and written in correct English. The authors properly describe both pre-clinical and clinical studies using Hericium erinaceus in AD related pathologies, by provide two clare and complete tables. Furthermore, they provide an elegant speculation on the mechanisms of action of monkey head mushroom considering several cellular and molecular aspects. I endhorse the pubblication of this review in its present form and I do not see any reason to add furher comments.

Author Response

Response to Reviewer #2

[General Comment] In this review Yanshree et colleagues report a wide view for the therapeutical use of  monkey head mushroom as memory enhancer on AD. The review is concise yet complete and written in correct English. The authors properly describe both pre-clinical and clinical studies using Hericium erinaceus in AD related pathologies, by provide two clare and complete tables. Furthermore, they provide an elegant speculation on the mechanisms of action of monkey head mushroom considering several cellular and molecular aspects. I endhorse the pubblication of this review in its present form and I do not see any reason to add furher comments.

Response: Thank you very much for your appreciation.